# Lack of Tradeoff between Leaf Hydraulic Efficiency and Safety across Six Contrasting Water-Stress Tolerant Fruit Tree Species

**Marco Isaac Garrido** [1,2,*] and **Sebastián Vergara** [2]

1 Centro de Estudios en Zonas Áridas (CEZA), Universidad de Chile, Km 45 Ruta D43, Coquimbo 1780000, Chile
2 Departamento de Producción Agrícola, Facultad de Ciencias Agronómicas, Universidad de Chile, Av. Santa Rosa 11315, La Pintana, Santiago 8820808, Chile
* Correspondence: marcogarrido@uchile.cl

**Abstract:** Water deficits affect the capacity of leaves to transport water, a process that is related to the obstruction of air in the xylem (embolism). The tolerance to this process has been negatively associated with water-transport efficiency at the xylem level across species, suggesting a tradeoff between hydraulic efficiency and safety. But there is a lack of observation at higher integration levels, i.e., organs. This study aimed to evaluate this tradeoff across six fruit tree species with a wide range of water-stress tolerance: pomegranate, olive, fig tree, mandarin, avocado, and vine. Efficiency was represented by the maximum foliar hydraulic conductance ($K_{max}$) and stomatal conductance, whereas hydraulic security by water potential in which the leaf loses 50% of its water-transport capacity ($P_{50}$), and at the point of loss of leaf turgor ($\Psi_{tlp}$). Results suggest that the compensation is weak or null at the foliar level. We observed that species with higher hydraulic efficiency tend to be more tolerant to leaf dehydration (higher hydraulic safety), except mandarin, which had lower $K_{max}$ and relatively higher $P_{50}$. Morphological traits associated with carbon investment dynamic (leaf mass per area and petiole density) were highly correlated to water-stress tolerance across fruit tree species.

**Keywords:** water-stress tolerance; hydraulics; leaf hydraulic conductance; $P_{50}$; turgor loss point; vulnerability curve; leaf mass per area; fruit trees



## 1. Introduction

Climate change effects are becoming more evident and threatening to the planet's biota every day. This threat tends to aggravate the hydrological scenario of zones with water deficits like drylands, because they belong to areas of high vulnerability to the phenomenon of increased frequency, magnitude, and duration of drought events, producing critical levels of availability of water [1]. The spatial and temporal variation of water availability potentially selects the hydraulic architecture of trees and can explain the distribution patterns of terrestrial plants [2]. This concept is defined by the water-transport system structure of plants [3], wherein the xylem is constituted as a specialized tissue for passive long-distance water transport. This hydraulic architecture determines the number of leaves that can be supplied with water [4], and it is related to the transverse and vertical segmentation of the xylem tissue, hydraulic conductance, xylem vulnerability to cavitation, tissue capacitance, and Huber value. These have a qualitative and quantitative character and are expressed at different integration levels, i.e., tissue, organ, and individual [3]. Other traits associated with hydraulic architecture are tree height, conductive tissue density, specific leaf area, and other anatomical traits. Each species exhibits a combination in its hydraulic architecture governed by biophysical rules that establish patterns in water transport [5], and efficient water-transport systems allow plants to sustain water evaporation from leaves and therefore, photosynthesis [6].

Leaves are the most plastic organ against environmental conditions [7]. It has been reported that leaf-blade traits reflect the effects of water stress more clearly than stem

and/or root traits [8]. Water transport efficiency in leaves, measured as leaf hydraulic conductance ($K_{leaf}$), is achieved by investing in large xylem vessels [9], with a high vein density and a shorter length of the extra-xylem pathway [10]. High efficiency is associated with higher water consumption and maximum net assimilation rate ($A_{max}$) because leaves can supply more water to the photosynthetic cells of the mesophyll [11]. Maire et al. [12] predicted that plants with higher hydraulic efficiency have high rates of maximum photosynthesis, high leaf element concentrations, and high leaf mass per area (LMA). A positive association between $A_{max}$ and $K_{leaf}$ [10], and stomatal conductance and plant hydraulic conductance [13] were observed, suggesting coordination between liquid- and gas-phase transport in plants [14].

In a water-deficit condition, sustaining carbon capture through photosynthesis and cell growth requires maintaining $K_{leaf}$ and leaf cell turgor [15]. As the leaf dehydrates, the tissue's water potential decreases, promoting a reduction of $K_{leaf}$ induced by cavitation and embolism [16]. It is postulated that greater stomatal regulation will occur in cavitation-vulnerable species, whereas more tolerant species will perform less stomatal regulation of water potential in a continuum iso-anisohydric spectrum, i.e., a lesser and more conservative stomatal regulation of water potential, respectively [17]. The function that results from the relationship between $K_{leaf}$ and leaf water potential is known as the leaf hydraulic vulnerability curve [18], which describes the loss of $K_{leaf}$ as a function of leaf water potential. Through the analysis of the vulnerability curve, critical points can be inferred such as the leaf water potential in which 12 ($P_{12}$), 50 ($P_{50}$), and 88% ($P_{88}$) of $K_{leaf}$ are lost [19]. These parameters allow species to be compared and have been related to functional traits of resistance to water stress at the foliar level, such as the iso-anisohydric behavior, water potential at turgor loss point ($\Psi_{tlp}$), and osmotic adjustment, among others [20]. $P_{50}$ provides information on the issue of where small changes in leaf water potential result in large changes in hydraulic conductance, which represents the inflection point of the vulnerability curve [21].

It has been postulated that the efficiency traits are compensated by hydraulic safety [22,23], i.e., smaller vessel diameter, increasing resistance to cavitation, and thicker walls that minimize the risk of vessel implosion at higher water tension, however, increasing resistance to water flow [24]. This theoretical tradeoff between hydraulic efficiency and safety, documented at the xylem tissue level [25,26], suggests that plant species sacrifice their water transport efficiency when they invest in greater safety. It has been proposed that this tradeoff is unavoidable at the level of individual membrane-pores, but this correlation could weaken as the level of analysis broadens to include whole membranes, whole conduits, and whole xylem tissue [27]. Nevertheless, a safety versus efficiency tradeoff was observed in C4 grass leaves [28], and, in neotropical canopy liana and tree species, as lianas are more efficient in transporting water, but more vulnerable to cavitation than trees [29].

Much has been discussed about the need to incorporate species and cultivars resistant (i.e., tolerance and avoidance; [7]) to water stress. To achieve this aim, it is necessary to understand the response mechanisms to the water deficit of currently cultivated species to optimize their management and predict the possibilities of establishment, survival, and distribution of species in the future [5]. Efficiency and hydraulic safety in species of agricultural interests have been limited to model species like grapevine [30], and less frequently in species like olive [31], but the study of those mechanisms should help to advance the identification of relevant physiological targets in the research of plant material more tolerant of and resilient to dryer conditions [32]. Furthermore, selected traits of water stress tolerance could be candidates for other abiotic stressors that have similar or common response mechanisms, like salinity stress [33,34]. Salinity is widely extensive as a stress factor in agricultural soils interacting with drought stress under a climate change scenario [35,36]. Thus, this research aimed to study the relationship between efficiency and hydraulic safety traits at the leaf level in six fruit tree species of different origins and therefore with contrasting water-stress tolerance. We hypothesize that the maximum

leaf hydraulic conductance is positively related to $P_{50}$ and $\Psi_{tlp}$, evidencing a hydraulic efficiency–safety tradeoff at the leaf level across fruit tree species.

## 2. Materials and Methods

### 2.1. Study Site Description

This study was carried out at the Centro de Estudios de Zonas Áridas of Universidad de Chile, located at the Las Cardas Experimental Station, Elqui Province, Coquimbo region. The geographical coordinates of the experimental field are 30°14′ south latitude, 71°14′ west longitude, and 260 m altitude. The bioclimate of the study area corresponds to the Desert–Ocean Mediterranean [37].

### 2.2. Biological Material

The study included six fruit tree species that are grown in the arid and semi-arid areas of northern Chile (Table 1). The one-year-old plants were obtained from a local nursery, and six plants of each species were transplanted in October 2019 in 20-L pots with a substrate composed of a 1:1 *v/v* mixture of peat (DMS2 Protekta; NPK of 15-12-29 + microelements in a concentration of 0.6 kg m$^{-3}$) and agricultural land. The pots have perforations at their base to allow free drainage and were put over platforms to avoid radical growth.

**Table 1.** Species, cultivars, and rootstocks were included in the trial.

| Scientific Name | Common Name | Cultivar |
| --- | --- | --- |
| *Citrus reticulata Blanco* | Mandarin | Orogrande (grafted over 'Carrizo' (*Citrus sinensis* L. Osb.× *Poncirus trifoliata* L. Raf.)) |
| *Ficus carica* L. | Fig tree | Black Mission |
| *Olea europaea* L. | Olive | Sevillana |
| *Persea americana Mill.* | Avocado | Hass (grafted over 'Mexícola' (*Persea americana* Mill.)) |
| *Punica granatum* L. | Pomegranate | Wonderfull |
| *Vitis vinifera* L. | Vine | Emperor |

### 2.3. Experimental Setup

The study was carried out between January and February 2020. Six fruit trees per species were arranged in a completely randomized experimental design with six replications. Trees were chosen to maximize homogeneity between experimental units, which was defined as a tree in a pot. Trees were acclimatized to the outdoor conditions between October and January. Plants were manually irrigated every 2–3 days to meet their water demand before and during the development of the experiment, no water deficits were implemented in this study. The substrate surface in each pot was covered with a layer of black Rachell mesh to minimize direct evaporation.

We use protocols of bench dehydration to infer water stress tolerance across species. That was made to generate a gradient of leaf water potential and relate it to leaf hydraulic conductance. Also, for measurement of parameters derived from volume-pressure curves.

### 2.4. Measurements and Estimates

#### 2.4.1. Leaf Hydraulic Conductance

Leaf hydraulic conductance ($K_{leaf}$; mmol s$^{-1}$ m$^{-2}$ MPa$^{-1}$) in function to leaf water potential was estimated through the partial rehydration method [38] based on the following equation:

$$K_{leaf} = C_{leaf} \frac{\ln\left(\frac{\Psi_o}{\Psi_f}\right)}{t},$$
(1)

where $C_{leaf}$ is the absolute leaf capacitance (mmol m$^{-2}$ MPa$^{-1}$), $\Psi_o$ and $\Psi_f$ are the leaf water potentials before and after partial rehydration (−MPa), respectively, and t is the rehydration time (s).

Two branches per experimental unit were sampled early in the morning (low xylem tension) and were quickly taken to the laboratory. To ensure correct hydration of the samples, branches were immediately trimmed underwater and left hydrating in opaque plastic bags for two hours. Then, samples were bench-dehydrated for a variable time to achieve a gradient of $\Psi_o$. Before each measurement, each branch was put in opaque plastic bags for 10–30 min to equilibrate, minimizing differences of water potential between leaves or twigs (pomegranate and olive) of the same branch. $\Psi_o$ was measured with a pressure chamber (model 1505D EXP, PMS Instrument Company, Albany, NY, USA). Immediately afterward, an adjacent leaf or twig was cut underwater leaving the cut submerged for 30 s (t; Equation (1)) to allow partial rehydration. After that, $\Psi_f$ was measured with the same pressure chamber. Each $K_{leaf}$ point was estimated with two leaf or twigs (between 24 and 36 leaves of twigs measured per species). This measurement was repeated during a bench dehydration period until the difference between $\Psi_o$ and $\Psi_f$ was minimal (<0.5 MPa), assuming a minimum leaf hydraulic conductance. $C_{leaf}$ (Equation (1)) was estimated from pressure–volume curves (PVC).

### 2.4.2. Pressure-Volume Curves (PVC) Traits

PVC were made on two fully expanded leaves or twigs per tree, which were sampled early in the morning and quickly taken to the laboratory. Samples were placed in a container with distilled water and left hydrated for two hours. Then, repeated measurements of fresh weight (FW) and leaf water potential were measured during a bench-dehydration period until the samples reached a potential of $-2.5$ MPa (avocado and vine) and $-4.0$ MPa (pomegranate, olive, fig tree, and mandarin). Full turgor weight (FTW) was estimated as the x-intercept of the linear relationship between the three or four first measured points of water potential and leaf/twig water content (when the relationship between both was linear). Finally, the leaf area was measured, and samples were dried in a forced-air oven at 60 °C to constant weight to obtain the value of dry weight (DW), used to calculate the relative water content (RWC; Equation (2)) as follows:

$$\text{RWC}: \frac{\text{FTW} - \text{FW}}{\text{FTW} - \text{DW}}. \tag{2}$$

PVC analysis was performed by plotting the relationship between $-1/\Psi$ and 1-RWC. Once the PVC was adjusted, it was possible to estimate water potential at the turgor loss point ($\Psi_{tlp}$; MPa), full turgor osmotic potential ($\Psi_o$; MPa), elasticity modulus ($\varepsilon$; $\text{MPa}^{-1}$), and leaf hydraulic capacitance before (TLC; $\text{moL m}^{-2} \text{ MPa}^{-1}$) and after (TLC*; $\text{mol m}^{-2} \text{ MPa}^{-1}$) turgor loss point [39].

### 2.4.3. Predawn ($\Psi_{pd}$), Xylem Midday ($\Psi_{xyl}$) Water Potential, and Stomatal Conductance ($g_s$)

$\Psi_{pd}$ ($-$MPa) was measured between 4:00 and 6:00 h, and $\Psi_{xyl}$ ($-$MPa) between 13:00 and 14:00 h. Both measurements were carried out between 30 January and 07 February 2020, and were performed on two leaves or branches per tree, on four trees per species with a model 1505D EXP pressure chamber (PMS Instrument Company, Albany, NY, USA) through a standard procedure [40]. $\Psi_{xyl}$ was measured on terminal leaves or twigs (pomegranate and olive) that were previously covered with plastic and aluminum to suppress transpiration and allow water potential equilibrium. $g_s$ was measured with a steady-state porometer (DECAGON Devices, Steady State Diffusion Leaf Porometer Model SC-1) in four sunny and fully expanded leaves per tree on the same days and in the same period in which $\Psi_{xyl}$ was measured.

### 2.4.4. Leaf Mass per Area (LMA) and Petiole Density ($D_p$)

Two fully expanded leaves per tree were sampled, and the leaf blade was separated from the petiole. Petiole volume was estimated through a geometrical method, measuring the length and basal and terminal diameter of the central section of each petiole or basal section of twigs in the case of pomegranate and olive. Leaf blades were scanned to

determine their area by using ImageJ software. Petioles and leaf blades were dried in a forced-air oven (Venticell, Grupo MMM, Planegg, Germany) at 60 °C until constant weight and their dry weight was measured on a precision balance. Petiole density was estimated as the ratio between dry mass and volume, whereas leaf mass per area was estimated as the ratio between leaf mass and its leaf area.

2.4.5. Leaf $^{13}$C Isotopic Composition ($\delta^{13}$C)

The $\delta^{13}$C measurements were made through standard protocol [26,41]. Leaves sampled to LMA and Dp measurement were crushed with a mortar until a fine powder, which was encapsulated in a tin capsule. The leaf $^{13}$C isotopic composition of each sample was measured at the Stable Isotope Laboratory of the Faculty of Agricultural Sciences of the Universidad de Chile with an isotopic ratio mass spectrometer (IRMS) model IMTEGRA2 (Sercon Ltd. Cheshire, Crewe, UK).

*2.5. Statistical Analysis*

Variables and parameters were analyzed through a one-way analysis of variance with species as a factor. When a significant factor effect was detected, means were discriminated through a post hoc DGC analysis with a significance level of 0.05.

Linear Pearson correlations were performed between traits and linear regression was made for relevant relations. Both had a significance level of 0.05.

Leaf Hydraulic Conductance Vulnerability Curves

In each species, a Weibull function [42] (Equation (3)) was adjusted to the relationship between leaf hydraulic conductance ($K_{leaf}$) and a leaf water potential, by using the fitplc R package [43]. Thus, water potential in which 12%, 50%, and 88% ($P_{12}$, $P_{50}$, and $P_{88}$, respectively) of the maximum leaf conductance ($K_{max}$) are lost was determined. Parameters were estimated through Bootstrap methods. $K_{max}$ was estimated through the average of the five highest $K_{leaf}$ values of each species,

$$K = K_{max}\left(1 - \frac{X}{100}\right)^{\left[\left(\frac{\Psi_{xyl}}{\Psi_x}\right)^{\frac{\Psi_x \, S_x}{V}}\right]}, \tag{3}$$

where $K_{max}$ is the maximum leaf hydraulic conductance, $\Psi_{xyl}$ is the xylem water potential, $\Psi_x$ is the xylem water potential where x% of the conductance is lost, $S_x$ is the slope of the curve at $\Psi_{xyl} = \Psi_x$, and V is a function setting parameter.

## 3. Results
### 3.1. Water Status of the Species

Table 2 shows conductances and water potential variables for six fruit tree species. The $g_s$ was significantly different between species ($p < 0.0001$), with mandarin being the species with the lowest $g_s$, and olive and fig tree being the species with the highest $g_s$. $K_{max}$ was significantly different between species ($p < 0.0001$). The $\Psi_{xyl}$ was significantly different between species ($p = 0.0033$), where olive and mandarin species had the most negative values at noon. $\Psi_{pd}$ did not differ significantly between the species ($p = 0.7145$).

### 3.2. Morpho-Physiological Traits Related to Water Relations

Table 3 shows integrative morpho-physiological traits across six fruit tree species. A significant effect of species was observed in LMA ($p < 0.0001$). Olive and grapevine had the highest and lowest values, respectively. Three statistically different groups were formed for $D_p$ ($p < 0.0001$), highlighting the mandarin and olive groups for exhibiting the highest values. Regarding the isotopic composition ($\delta^{13}$C), two statistically different groups were obtained ($p = 0.0071$), wherein the group of fig, avocado, and grapevine had the most negative values (Table 3).

**Table 2.** Stomatal conductance ($g_s$), maximum leaf hydraulic conductance ($K_{max}$), predawn water potential ($\Psi_{pd}$), and xylem water potential ($\Psi_{xyl}$) of six fruit species.

| Species | $g_s$ (mmol m$^2$ s$^{-1}$) | $K_{max}$ (mmol MPa$^{-1}$ m$^2$ s$^{-1}$) | $\Psi_{pd}$ (MPa) | $\Psi_{xyl}$ (MPa) |
|---|---|---|---|---|
| Avocado | 262.0 ± 48.7 b | 16.9 ± 0.77 b | −0.48 ± 0.02 a | −1.08 ± 0.10 a |
| Fig tree | 616.7 ± 61.6 a | 23.5 ± 1.65 a | −0.54 ± 0.09 a | −1.28 ± 0.12 a |
| Mandarin | 145.3 ± 51.1 c | 8.0 ± 0.34 c | −0.59 ± 0.06 a | −1.52 ± 0.09 b |
| Olive | 601.9 ± 45.9 a | 21.1 ± 0.90 a | −0.61 ± 0.05 a | −1.61 ± 0.15 b |
| Pomegranat | 380.6 ± 52.1 b | 19.1 ± 0.72 b | −0.57 ± 0.04 a | −1.31 ± 0.10 a |
| Vine | 353.6 ± 52.1 b | 9.0 ± 0.26 c | −0.53 ± 0.07 a | −1.04 ± 0.10 a |

Means ± standard error (n = 6). Values with the same letter in a vertical way do not differ significantly according to the post hoc DGC analysis ($p > 0.05$). $K_{max}$ were compared through 95% bootstrap confidence intervals.

**Table 3.** Leaf mass per area (LMA), petiole density ($D_p$), and leaf $^{13}$C isotopic composition ($\delta^{13}$C) of six fruit tree species.

| Species | LMA (g m$^{-2}$) | $D_p$ (mg mm$^{-3}$) | $\delta^{13}$C (‰) |
|---|---|---|---|
| Avocado | 108.5 ± 4.1 b | 0.32 ± 0.01 b | −27.5 ± 0.6 b |
| Fig tree | 86.7 ± 5.6 c | 0.21 ± 0.02 a | −27.8 ± 0.6 b |
| Mandarin | 124.3 ± 10.5 b | 0.44 ± 0.02 c | −25.1 ± 0.6 a |
| Olive | 181.0 ± 14.4 a | 0.53 ± 0.05 c | −25.6 ± 0.3 a |
| Pomegranate | 95.8 ± 3.0 c | 0.35 ± 0.06 b | −25.3 ± 0.3 a |
| Vine | 52.9 ± 4.1 d | 0.14 ± 0.01 a | −26.5 ± 0.8 b |

Mean ± standard error (n = 6). Values with the same letter in a vertical way do not differ significantly according to the post hoc DGC analysis ($p > 0.05$).

### 3.3. Pressure–Volume Curve Traits Curve

Table 4 shows that the vine had a significantly higher $\Psi_o$ (−1.13 MPa) than the rest of the species. Concerning $\Psi_{tlp}$, statistically significant differences were observed ($p < 0.0001$), highlighting again the vine for its higher $\Psi_{tlp}$, followed by avocado and fig tree. Olive, pomegranate, and mandarin were species with the lowest $\Psi_{tlp}$. The parameters $\varepsilon$, TLC, and TLC* were statistically different between species ($p = 0.0116$; $p = 0.0039$ and $p = 0.0185$, respectively); avocado and grapevine were the ones with the highest modulus of elasticity, and lower leaf capacitance (Table A2).

**Table 4.** Osmotic potential at full turgor ($\Psi_o$) and water potential at turgor loss point ($\Psi_{tlp}$) of six fruit tree species.

| Species | $\Psi_o$ (MPa) | $\Psi_{tlp}$ (MPa) |
|---|---|---|
| Avocado | −1.53 ± 0.09 b | −1.85 ± 0.07 b |
| Fig tree | −1.64 ± 0.19 b | −2.10 ± 0.19 b |
| Mandarin | −1.66 ± 0.12 b | −2.33 ± 0.15 c |
| Olive | −1.86 ± 0.23 b | −2.75 ± 0.08 c |
| Pomegranate | −1.81 ± 0.22 b | −2.48 ± 0.18 c |
| Vine | −1.13 ± 0.03 a | −1.44 ± 0.01 a |

Means ± standard error (n = 6). Values with the same letter in a vertical way do not differ significantly according to the post hoc DGC analysis ($p > 0.05$).

### 3.4. Leaf Hydraulic Conductance Vulnerability

Figure 1 shows vulnerability curves for leaf hydraulic conductance as a relationship between percentage loss of leaf conductance (PLC) and leaf water potential (in positive values), and Table 5 shows $P_{50}$ and $S_x$ parameter values and their respective bootstrap 95% confidence intervals. Avocado was the species with the highest $P_{50}$, being more hydraulically vulnerable. The vine, fig tree, and mandarin had lower $P_{50}$ than avocado, followed by olive and pomegranate, the species with the lowest $P_{50}$ (Table 5). Pomegranate hadlower $S_x$, i.e., the smallest increase in the percentage loss of hydraulic conductance per unit of decrease in leaf water potential, similar to olive. Mandarin and fig tree had similar $S_x$, and vine and avocado had lower $S_x$.

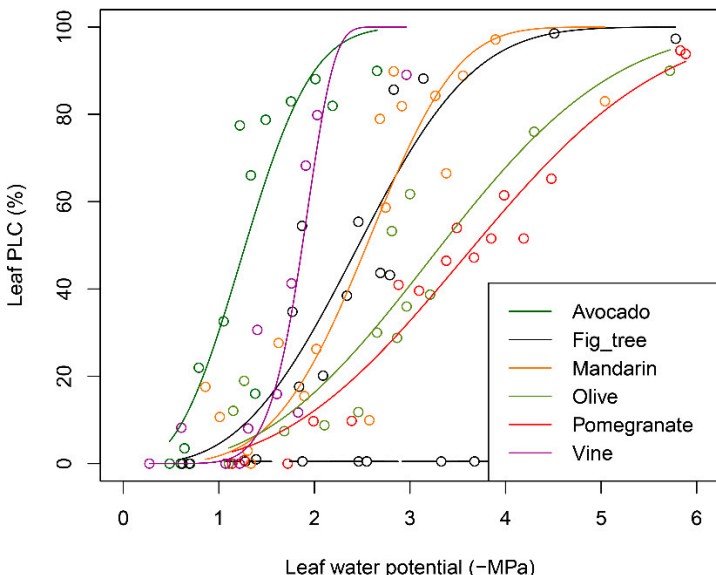

**Figure 1.** Leaf hydraulic conductance vulnerability curves for six fruit tree species. Black dots and horizontal bars at the bottom of the graph represent the mean and bootstrap 95% confidence interval of $P_{50}$.

**Table 5.** Water potential in which leaf loss was 50% of hydraulic conductance ($P_{50}$) and slope of the relationship between leaf hydraulic conductance and leaf water potential ($S_x$) for six fruit tree species.

| Species | $P_{50}$ | Boot–2.5% | Boot–97.5% | $S_x$ | Boot–2.5% | Boot–97.5% |
|---|---|---|---|---|---|---|
| | (MPa) | | | $(MPa^{-1})$ | | |
| Avocado | −1.28 a | 1.07 | 1.55 | 72.44 a | 54.13 | 210.34 |
| Fig tree | −2.46 b | 2.16 | 2.73 | 42.26 a | 25.33 | 74.63 |
| Mandarin | −2.55 b | 2.29 | 2.88 | 53.18 a | 38.01 | 844.06 |
| Olive | −3.33 c | 2.92 | 3.68 | 27.84 a | 18.48 | 84.29 |
| Pomegranate | −3.68 c | 3.50 | 3.85 | 26.00 a | 21.39 | 31.66 |
| Vine | −1.87 b | 1.74 | 2.31 | 145.75 a | 43.37 | 947.41 |

Boot−2.5% and−97% refer to bootstrap 90% confidence intervals of values with the same letter in a vertical way do not differ significantly according to the comparison between the 95% bootstrap confidence intervals.

### 3.5. Relationship between Hydraulic, Water Stress Tolerance, and Morphological Traits

The relationship between efficiency and safety hydraulic parameters was evaluated across species (Figure 2a,b). No significant association was observed between hydraulic security traits $\Psi_{tlp}$ ($p = 0.349$), and $P_{50}$ ($p = 0.468$) with $K_{max}$ as a hydraulic efficiency trait. A positive and significant association was observed between $g_s$ and $K_{max}$ ($p = 0.00072$) (Figure 2c).

Midday xylem water potential was significant and positively related to $\Psi_{tlp}$ ($p = 0.015$). It was observed that in all species the minimum daily potential always remained above the water potential at the turgor loss point, being the slope of the relationship 1.84 MPa MPa$^{-1}$ (Figure 3a). Midday xylem water potential was significant and negatively related to LMA ($p = 0.041$; Figure 3b) and $D_p$ ($p = 0.030$; Figure 3c).

### 3.6. Correlation Analysis

Associations between traits are represented through a correlation matrix (Figure 4), where both positive and negative significant correlations were highlighted as blue and red filled boxes, respectively. Leaf water-stress tolerance trait $\Psi_{tlp}$ was positively associated with $P_{50}$ ($p = 0.047$) and $S_x$ ($p = 0.012$). $P_{50}$ and $P_{88}$ were positively associated ($p = 0.0004$). Morpho-anatomical traits LMA and $D_p$ were positively associated among them ($p = 0.004$), and both were negatively associated with $\Psi_{tlp}$ ($p = 0.043$ and $p = 0.026$, respectively). $\Psi_{tlp}$ and $\Psi_o$ were positively associated ($p = 0.0024$).

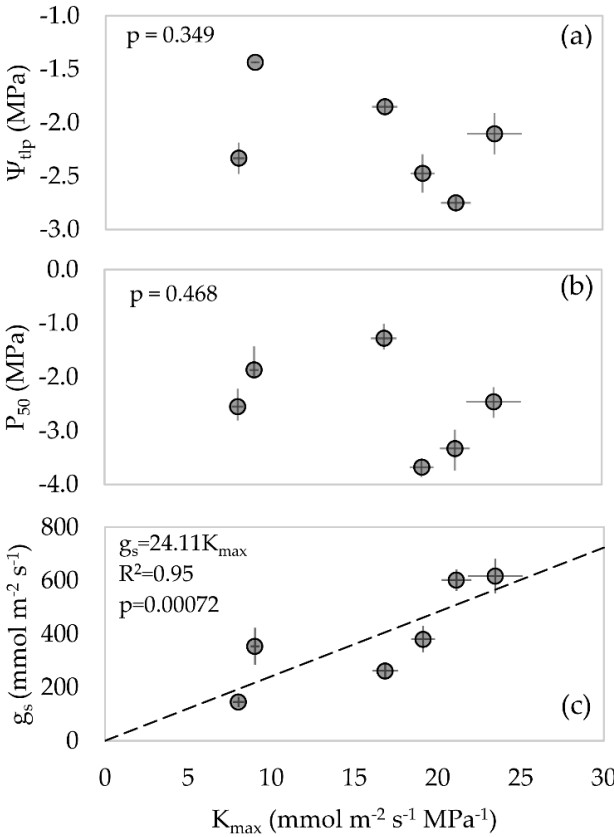

**Figure 2.** Relationship of $K_{max}$ with $\Psi_{tlp}$ (**a**), $P_{50}$ (**b**), and $g_s$ (**c**). Each point represents the mean per species $\pm$ standard error (n = 6). Vertical bars in Figure 3b represent the 95% confidence interval for $P_{50}$.

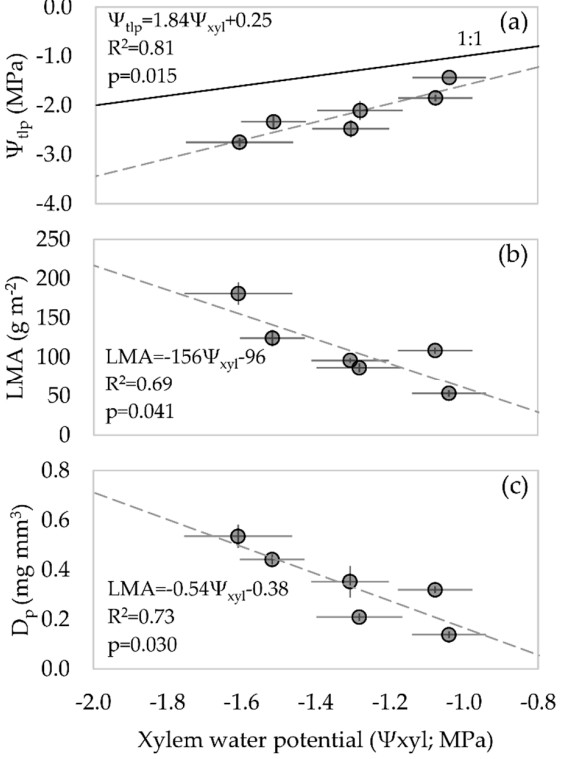

**Figure 3.** Relationship of $\Psi_{xyl}$ with $\Psi_{tlp}$ (**a**), LMA (**b**), and $D_P$ (**c**). Each point represents the mean per species $\pm$ standard error (n = 6). The solid line in (**a**) represents a 1:1 relationship between $\Psi_{xyl}$ and $\Psi_{tlp}$.

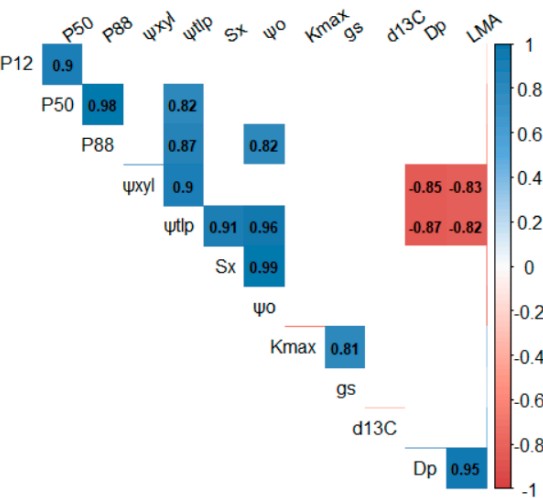

**Figure 4.** Correlogram. The value in each interaction box represents the Pearson correlation coefficient. Only significant associations ($p < 0.05$; $n = 6$) are shown. Red and blue colors indicate whether the association is negative or positive, respectively, whereas the color intensity indicates the degree of significance. Variables: stomatal conductance ($g_s$), xylem water potential ($\Psi_{xyl}$), leaf mass per area (LMA), petiole density ($D_p$), $^{13}C$ leaf isotopic composition ($d^{13}C$), water potential at turgor loss point ($\Psi_{tlp}$), osmotic potential at full turgor ($\Psi_o$), maximum leaf hydraulic conductance ($K_{max}$), leaf water potential in which 12 ($P_{12}$), 50 ($P_{50}$) and 88% ($P_{88}$) of the leaf hydraulic conductance are lost, and slope of the vulnerability curve ($S_x$).

## 4. Discussion

### 4.1. No Tradeoff between Efficiency and Hydraulic Safety across Six Fruit Tree Species

The hydraulic vulnerability curves (Figure 1) allow for verifying the hydraulic strategies deployed by the species in the face of water stress, which might be related to the degree of a tradeoff between efficiency and hydraulic safety [44]. Despite plants benefitting from both efficient and safe leaves in terms of avoiding harmful xylem tension as shown in the Whitehead–Jarvis water-transport model [45], we observed that the most vulnerable species (less hydraulic security), avocado and vine (see $P_{50}$; Table 5), showed lower $K_{max}$ than Pomegranate and Olive, but higher than Mandarin (Table 2). These results agree with those reported by Gleason et al. [27] and Liu et al. [44], who observed that a large percentage of studied species exhibited low efficiency and hydraulic safety, suggesting that a tradeoff between these two hydraulic traits would not be widespread. Gleason et al. [27] indicate that understanding the drivers of efficiency and their specific tradeoffs with safety, as well as other functional traits, is necessary to understand hydraulic strategies. Similarly, Westoby and Wright [46] report that the species tend to be outside the optimal zone of compensation (high efficiency and hydraulic safety) because resistances in the vessel lumen will have been coordinated by natural selection with resistances passing through walls between vessels. The results obtained in this research suggest that, through the fruit tree species, there would be no hydraulic compensation at the foliar level between efficiency and safety (Figure 2). At higher levels of integration, i.e., leaves, many processes can alleviate the water stress, supporting a higher hydraulic efficiency at the tree level. That is the case of drought-induced osmotic adjustment that may influence a hydraulic efficiency-safety tradeoff [44], especially when water flow follows a xylem and extra-xylem path, like in leaves. Also, leaf shedding may maximize the sapwood area:leaf area ratio as a strategy to avoid water deficit allowing plants rising their leaf water potential even under drought [47] protecting high carbon cost organs. From an intensive point of view, Pomegranate and Olive deployed a tolerance strategy, with a $P_{50}$ of $-3.7$ MPa and $-3.3$ MPa, significantly lower than the rest of the species. However, those species showed higher maximum leaf hydraulic conductance (Table 2). This result has an ecophysiological sense because species adapted to Mediterranean and arid environments would require high rates of water transport to

keep the leaves hydrated and fully functional, necessitating an efficient water-transport system [48,49], and need fast water use following rain to minimize drainage and run-off in proximal events [50].

Regarding this, Gleason et al. [27] suggest that the hypothetical negative correlation between safety and efficiency may be unavoidable at low integration levels (e.g., xylem), but this correlation weakens at higher integration levels.

### 4.2. Ranking of Tolerance to Water Stress and Association between Tolerance Traits and Foliar Morphology

It is established that greater investment in hydraulic safety allows plants to operate at higher stresses and with fewer gas blockages within the xylem conduits [27]. This means that the species must exhibit more negative $P_{50}$ values [51]. Species constituted three groups in a gradient of hydraulic safety: pomegranate and olive in the high tolerance range, followed by fig and mandarin in the medium tolerance range, and finally vine and avocado in a low tolerance range. However, given the contribution of the extra-xylem pathway to leaf hydraulic conductance [10,52], the $\Psi_{tlp}$ may complement this characterization. In our study, intensive traits related to water stress tolerance $P_{50}$ and $\Psi_{tlp}$ were positively related ($p = 0.047$), as observed by Bartlett et al. [20]. Meinzer et al. [53] proposed $\Psi_{tlp}$ as an indicator of the degree of iso-anisohydrism, because it integrates traits associated with the ranges of stomatal control over water potential.

$\Psi_{tlp}$ has been associated with osmoregulation, which enables leaves to maintain turgor pressure under stress conditions associated with water availability like drought [20] or salinity stress [54]. Nevertheless, it is important to advance in the studies of osmotic function to deal with water and salinity stress, because differential responses in terms of osmoregulation depend on the type of the stress, i.e., salinity or water stress, have been observed [55,56].

Johnson et al. [15] observed that more anisohydric species, i.e., that experience the lowest and most variable water potential during the season, have lower values of $\Psi_{tlp}$, in addition to lower values of LMA. They also observed that these species were able to adjust these traits, investing in security when the intensity of the water deficit increased in the season. In our study, we observed an association between LMA and $D_p$, and both were associated with $\Psi_{tlp}$. Moreover, species that exhibit higher LMA and $D_p$, and lower $\Psi_{tlp}$ experienced the lowest $\Psi_{xyl}$ at noon, with the same $\Psi_{pd}$ (equal water availability). Thus, a link between morphological traits (mechanical) and hydraulic safety across species is established. This would be important to maintain growth and stomatal opening in the face of water stress [39]. This coincides with those exposed by McCulloh et al. [13], who postulate that the mechanical requirements to tolerate high stresses within the xylem conducts, require a higher density of the tissues, which would translate into greater mechanical resistance. This greater tolerance to low water potentials would have an ecological consequence since an investment in higher-density tissues would imply slower growth rates [57] and higher carbon costs. It is necessary to notice that two of six species were grafted. Mandarin over "Carrizo" and Avocado over "Mexícola" (Table 1). It has been observed that the rootstock could modulate physiological and growth traits of the scion [58], and in consequence their tolerance to water deficit. Therefore, the cultivars may study have different behavior or ranking if they were own-rooted.

### 4.3. Leaf Hydraulic Function Ranges

The correlation between $\Psi_{tlp}$ and $\Psi_o$ has been reported in other studies [53]. Although all species experienced minimum xylem water potentials under irrigation always above $\Psi_{tlp}$, the slope of the relationship had a value of 1.84 MPa MPa$^{-1}$, which implies that the species with lower $\Psi_{tlp}$ had a safety margin higher in this condition. Those results can be interpreted in terms of hydraulic safety margins (HSM$_{\Psi x-P50}$ or HSM$_{\Psi x-\Psi tlp}$; both correlated at 0.01 of significance; r = 0.74; $p = 0.095$; $n = 6$). HSM$_{\Psi x-P50}$ in avocado shows a more restrictive operation range (0.2 MPa), followed by vine (0.83 MPa). Mandarin and

fig-tree showed similar $HSM_{\Psi x-P50}$ (1.03 and 1.18 MPa, respectively), whereas olive and pomegranate had the largest $HSM_{\Psi x-P50}$ (1.72 and 2.37 MPa, respectively). Species with the largest $HSM_{\Psi x-P50}$, as a minor $\Psi_{tlp}$ can be classified as more anisohydric [53,59] and could be good candidates for restrictive water management.

It was observed that all the species, in an irrigated condition, at noon reached xylem water potential very close to $P_{12}$ (Table A1), with possible exposure to slight degrees of cavitation. This was observed by Manzoni et al. [49] who indicated that in an irrigated condition the stomata would allow a degree of cavitation to reach a maximum capacity for transporting water and gas exchange. Furthermore, we observed an association between $K_{max}$ and $g_s$ (Figure 3c), which agrees with research that reported correlations between $g_s$, maximum net assimilation, and leaf hydraulic conductance ($K_{leaf}$) [38,47], and with a positive association between maximum transpiration rate and water stress tolerance traits in *Vitis vinifera* reported by Dayer et al. [30]. Nevertheless, the evaluation of tolerance to water deficits studied through intensive (independent of size) traits such as $\Psi_{tlp}$ and $P_{50}$ could be incomplete if traits associated with time to stress arrival or "stress distance" are not considered [30,60]. For example, a prioritization of radical growth that allows for avoiding water deficits through access to new water sources [61,62], or drought-induced leaf shedding that prevents rapid use of available water [47,63]. These adaptive traits could even determine the ability of the plant to acclimatize to a water-deficit condition, giving it enough time to modify morpho-physiological traits such as $P_{50}$ or $\Psi_{tlp}$ [64], improving its performance under a water deficit.

The coordination between liquid- and gas-phase transport in plants is also supported by coordinated changes across species in the water potential at stomatal closure and incipient cavitation [4,65]. During dry periods, a tightly stomatal regulation of water losses corresponds to a first hypothesis where leaf xylem conductance explains most of $K_{leaf}$ variation and stomatal and mesophyll conductance decreases during the progression of a water deficit, avoiding loss of leaf conductance [66]. Nevertheless, a second hypothesis is that leaf hydraulic conductance outside the xylem control water fluxes by acting as a hydraulic valve that reduces transpiration, being a driver of stomatal and mesophyll conductance [14,66]. If we assume coordination between $\Psi_{tlp}$ and the water potential at stomatal closure [59], in that case, we can test these hypotheses or strategies through the difference between the water potential where start the loss of leaf hydraulic conductance ($P_{12}$) and $\Psi_{tlp}$, being negative or near to zero differences associated to the first hypothesis and greater positive values to the second hypothesis. In our study, Vine and Pomegranate show differences of 0 and 0.48 MPa, respectively, while fig tree and Mandarine show differences of ~0.7 MPa, and Olive and Avocado of ~1 MPa. However, the linkages of $K_{leaf}$ and gas exchange are still under debate, and it recognizes the need for investigation to improve our knowledge about the relationship between these traits, productivity related to assimilation rate, and survivor capacity related to hydraulic failure [66].

## 5. Conclusions

The present study evaluates a hydraulic efficiency-safety tradeoff at leaf level across six fruit tree species. Maximum leaf hydraulic conductance, i.e., hydraulic efficiency, was not associated with leaf $P_{50}$ and $\Psi_{tlp}$, i.e., hydraulic safety, suggesting that a theoretical clear tradeoff at a low integration level, i.e., xylem, is not so clear at higher integration levels. In terms of water-stress tolerance level, we observed tree groups: pomegranate and olive in the high tolerance range, followed by fig and mandarin in the medium tolerance range, and finally vine and avocado in a low-tolerance range. More water-stress tolerant species show leaf morphological traits associated with higher carbon investments, i.e., denser tissue, with more carbon per unit of leaf area or petiole volume, suggesting an association between water-stress tolerance and tissue mechanical resistance. As observed in other studies, LMA and water transport tissue density could be good proxies for intensive traits of water stress tolerance, more easily and efficiently to measure than functional traits.

**Author Contributions:** Conceptualization, M.I.G.; methodology, M.I.G.; formal analysis, M.I.G. and S.V.; investigation, M.I.G. and S.V.; resources, M.I.G.; data curation, M.I.G. and S.V.; writing—original draft preparation, M.I.G.; writing—review and editing, M.I.G. and S.V.; supervision, M.I.G.; project administration, M.I.G.; funding acquisition, M.I.G. All authors have read and agreed to the published version of the manuscript.

**Funding:** This research was funded by Agencia Nacional de Investigación y Desarrollo (ANID), Chile, FONDECYT de Iniciación grant number 11190174 and Vicerrectoría de Investigación y Desarrollo, Universidad de Chile, Proyecto U-Inicia UI–015/19.

**Data Availability Statement:** The datasets generated for this study are available on request to the corresponding author.

**Conflicts of Interest:** The authors declare no conflict of interest.

## Appendix A

**Table A1.** Leaf hydraulic vulnerability curve parameter.

| Species | $P_{12}$ (MPa) | Boot-2.5% | Boot–97.5% | $P_{88}$ (MPa) | Boot–2.5% | Boot–97.5% |
| --- | --- | --- | --- | --- | --- | --- |
| Avocado | 0.68 | 0.5 | 1.08 | 1.94 | 1.32 | 2.34 |
| Fig_tree | 1.4 | 0.94 | 1.86 | 3.57 | 3 | 4.71 |
| Mandarin | 1.65 | 1.26 | 2.59 | 3.39 | 2.7 | 3.85 |
| Olive | 1.77 | 1.32 | 2.33 | 5.06 | 3.41 | NA |
| Pomegranate | 1.99 | 1.71 | 2.26 | 5.51 | 5.03 | NA |
| Vine | 1.51 | 1.12 | 1.83 | 2.16 | 1.93 | NA |

Boot−2.5% and−97% refer to bootstrap 90% confidence intervals of values.

## Appendix B

**Table A2.** Volume-Pressure curve parameter.

| Species | $\varepsilon$ (MPa$^{-1}$) | TLC (mol m$^{-2}$ MPa$^{-1}$) | TLC* (mol m$^{-2}$ MPa$^{-1}$) |
| --- | --- | --- | --- |
| Avocado | 19.77 ± 3.38 a | 0.67 ± 0.13 b | 1.07 ± 0.14 b |
| Fig tree | 10.19 ± 0.84 b | 1.41 ± 0.20 a | 2.43 ± 0.60 a |
| Mandarin | 12.67 ± 2.21 b | 1.17 ± 0.25 a | 2.73 ± 0.52 a |
| Olive | 11.80 ± 2.46 b | 1.22 ± 0.16 a | 2.90 ± 0.74 a |
| Pomegranate | 10.12 ± 1.74 b | 0.98 ± 0.15 a | 1.85 ± 0.53 a |
| Vine | 15.75 ± 1.51 a | 0.49 ± 0.04 b | 0.70 ± 0.08 b |

Mean ± standard error (n = 6). Values with the same letter in a vertical way do not differ significantly according to the post hoc DGC analysis ($p > 0.05$).

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
