# Peer review of "Lack of Tradeoff between Leaf Hydraulic Efficiency and Safety across Six Contrasting Water-Stress Tolerant Fruit Tree Species"

_agronomy, doi:10.3390/agronomy12102351_

Round 1

Reviewer 1 Report

Climate change effects are becoming more evident and severely affecting the ecological system. Therefore, the study area is more relevant and important. The authors have reasonably selected the techniques/parameters for evaluating the results relating to the objective of the study. However, I would like to provide the following comments and observations concerning this paper. 

  1. Lines 2-4 –  Title - suggest changing the title - to concise and
  2. Lines 10-21 – Abstract – suggest reorganizing the abstract, so that it is more specific and clearly states why, how, what, etc.  
  3. Line 14, 85 – selection of six fruit species over the others is not clear.
  4. Line 103, 112 – ‘ten plants ….’ – however, all the other placed six replicates. Is this correct?
  5. Line 155 – ‘RWC’ – This abbreviation has not been defined before.
  6. Lines 222-223 – state this as a note below the table. Follow the same for other tables too.
  7. Line 220. 233, 247 (Table 2, 3, 4)– use a period rather than a comma to indicate decimal places.
  8. Lines 243 – This abbreviation has not been defined before.
  9. Lines 259 – ‘…de lowes….’ Pls. check.
  10. Lines 277 – (Figure 3a y 3b) … pls. check.
  11. Line 279 – (Figure 4c) – I think this should be Figure 3c??
  12. Line 300 – (Figure 6) -- this should be Figure 5??
  13. Line 302-303 – P50 y P88 …- what is ‘y’ here? pls. check. GY – pls. explain??
  14. Line 322 – (table 2 and 5) – make ‘t’ upper case in the table.
  15. Line 337 – (e.g. Xilem.. – check the spelling here.
  16. Lines 339 – ..’(Figuera 4.B) – check spelling and figure label and make necessary changes.
  17. Lines 340 – …’gs’… check and make necessary changes.
  18. Line 352, 362 – [36] …. State the author's name at the start of the sentence. Follow the reference requirement in the journal and make all changes.
  19. Line 421-431 – delete all unnecessary texts/sections in the template.
  20. In-text references – check and make necessary changes to all in-text references to match with the ACS referencing style.

Author Response

Response to Reviewer 1

Climate change effects are becoming more evident and severely affecting the ecological system. Therefore, the study area is more relevant and important. The authors have reasonably selected the techniques/parameters for evaluating the results relating to the objective of the study. However, I would like to provide the following comments and observations concerning this paper. 

Reply: Thank you very much for all your observations. It helped us to improve the manuscript.

  1. Lines 2-4 –  Title - suggest changing the title - to concise and

Reply: Thank you for the suggestion. We modify the title to a more concise one: Lack of trade-off between efficiency and hydraulic safety at leaf level across six contrasting water stress tolerance fruit tree species

  1. Lines 10-21 – Abstract – suggest reorganizing the abstract, so that it is more specific and clearly states why, how, what, etc.  

Reply: Thank you for your observation. We reorganized the abstract. We hope that it is more clearly read now

  1. Line 14, 85 – selection of six fruit species over the others is not clear.

Reply: we select species to generate a water stress tolerance gradient. Thank you for your observation, we indicated that in the manuscript.

  1. Line 103, 112 – ‘ten plants ….’ – however, all the other placed six replicates. Is this correct?

Reply: We use six plants for species, thank you for noting this mistake.

  1. Line 155 – ‘RWC’ – This abbreviation has not been defined before.

Reply: we corrected it in the manuscript

  1. Lines 222-223 – state this as a note below the table. Follow the same for other tables too.

Reply: thank you for your suggestion. We make the changes in all tables of the manuscript.

  1. Line 220. 233, 247 (Table 2, 3, 4)– use a period rather than a comma to indicate decimal places.

Reply: we corrected it in the manuscript

  1. Lines 243 – This abbreviation has not been defined before.

Reply: you are right. We defined it in the materials and methods section.

  1. Lines 259 – ‘…de lowes….’ Pls. check.

Reply: we correct the word (the lower)

  1. Lines 277 – (Figure 3a y 3b) … pls. check.
  2. Line 279 – (Figure 4c) – I think this should be Figure 3c??
  3. Line 300 – (Figure 6) -- this should be Figure 5??

Reply to observations 10, 11, and 12: Thank you for your observation. We correct it in the manuscript: (Figure 3a and 3b), (Figure 3c), (Figure 5)

  1. Line 302-303 – P50 y P88 …- what is ‘y’ here? pls. check. GY – pls. explain??

Reply: we corrected it in the manuscript.

  1. Line 322 – (table 2 and 5) – make ‘t’ upper case in the table.

Reply: thank you for the observation. We corrected it in the manuscript.

  1. Line 337 – (e.g. Xilem.. – check the spelling here.

Reply: thank you for the observation. We corrected it in the manuscript.

  1. Lines 339 – ..’(Figuera 4.B) – check spelling and figure label and make necessary changes.

Reply: thank you for the observation. We corrected it in the manuscript (Figure 3c), and we check for other typos.

  1. Lines 340 – …’gs’… check and make necessary changes.

Reply: thank you for the observation. We corrected it in the manuscript

  1. Line 352, 362 – [36] …. State the author's name at the start of the sentence. Follow the reference requirement in the journal and make all changes.

Reply: thank you for your suggestion. We corrected it in the manuscript, and check other mistakes.

  1. Line 421-431 – delete all unnecessary texts/sections in the template.

Reply: thank you for your suggestion. We delete unnecessary sections.

  1. In-text references – check and make necessary changes to all in-text references to match with the ACS referencing style.

Reply: Thanks for your suggestion. We revised the manuscript to make sure there are no stylistic errors

Reviewer 2 Report

My Comments can be found in the attached PDF.

Author Response

Response to reviewer 2

First, thank you for your suggestions and literature recommendations. We appreciate your orientation and consider that some of the articles suggested were very important to improve the literature review in the manuscript.

We make a more comprehensive literature review that, I hope, improves the manuscript in terms of novelty.

Also, we improve the presentation of the results (figures and tables).

Round 2

Reviewer 2 Report

The Authors have addressed some of my comments but I am not convinced by the fact that they did not address point 2 and 3 (see below) of my previous review. I would recommend publication of the manuscript if those comments were more carefully addressed or if the Authors provided a detailed explanation for not doing so.

2. Role of other morphological adaptations. The role (or possible role) of morphological adaptations to water stress should be at least mentioned. Plants growing under water stress, for example, tend to invest in the belowground biomass at the expense of canopy growth (Friedlingstein, Joel et al. 1999, Bacelar, Moutinho-Pereira et al. 2007).

3. Role of other sources of stress. The role of other sources of abiotic stress rather than water stress is not even mentioned. For example, there is growing evidence and concern about the importance of salt stress (Perri, Suweis et al. 2018, Hassani, Azapagic et al. 2020), which could substantially affect plants’ gas exchange and lead to leaf anatomy adjustment similar to those observed in response to water stress (Huang, Li et al. 2019, Perri, Katul et al. 2019). I believe this should be mentioned. Salt-tolerant plants display an evident trade-off between efficiency and safety. Some have argued that it could be the result of osmoregulation. I believe these considerations should be made, and the manuscript would gain a broader perspective on the issues raised.

Author Response

Dear reviewer 1

Thank you again for your comments, they have helped us to substantially improve the manuscript that we have submitted for publication. Below you will find our answers.

Also, we attach the last version of the manuscript where you can find the modification whit the function of track changes.

We remain attentive to your reactions.

---

The Authors have addressed some of my comments but I am not convinced by the fact that they did not address point 2 and 3 (see below) of my previous review. I would recommend publication of the manuscript if those comments were more carefully addressed or if the Authors provided a detailed explanation for not doing so.

2. Role of other morphological adaptations. The role (or possible role) of morphological adaptations to water stress should be at least mentioned. Plants growing under water stress, for example, tend to invest in the belowground biomass at the expense of canopy growth (Friedlingstein, Joel et al. 1999, Bacelar, Moutinho-Pereira et al. 2007).

Thanks for your suggestion. We had been left with the idea that we had considered your comment in the latest version of the manuscript. For this reason, we reinforce it in the discussion by including new bibliographical references related to morpho-physiological traits with a role in a water stress condition. You can review the text included in the manuscript below.

Discussion section:

“This suggests that the evaluation of tolerance to water deficit studied through intensive (independent of size) traits such as Ψtlp and P50 could be incomplete if traits associated with time to stress arrival or "stress distance" are not considered [55,29]. e.g. a prioritization of radical growth that allows avoiding water deficit through access to new water sources [56, 57], or drought-induced leaf shedding that prevents rapid use of available water [58,59]. These adaptive traits could even determine the ability of the plant to acclimatize to a water deficit condition, giving it enough time to modify morpho-physiological traits such as P50 or Ψtlp [60], improving its performance under water deficit.”

3. Role of other sources of stress. The role of other sources of abiotic stress rather than water stress is not even mentioned. For example, there is growing evidence and concern about the importance of salt stress (Perri, Suweis et al. 2018, Hassani, Azapagic et al. 2020), which could substantially affect plants’ gas exchange and lead to leaf anatomy adjustment similar to those observed in response to water stress (Huang, Li et al. 2019, Perri, Katul et al. 2019). I believe this should be mentioned. Salt-tolerant plants display an evident trade-off between efficiency and safety. Some have argued that it could be the result of osmoregulation. I believe these considerations should be made, and the manuscript would gain a broader perspective on the issues raised.

R: Thank you very much for insisting on the comment. Initially, we thought this could be considered a deviation from the article's focus. However, after discussing it, we think that from a perspective it is important to discuss the possible role of water stress resistance traits when the stress is saline, or when there is an interaction between them. We have included information and bibliographic references in the introduction and discussion. As you suggest, we think that the manuscript gained a broader perspective. You can review the text included in the manuscript below.

Introduction section:

“Furthermore, selected traits to water stress tolerance could be candidates for other abiotic stress that have similar or common response mechanisms, like salinity stress [31,32], being salinity widely extended as stress factors in agricultural soils interacting with drought stress under a climate change scenario [33,34].”

Discussion section:

“Ψtlp has been associated with osmoregulation, which enables leaves to maintain turgor pressure under stress conditions associated with water availability like drought [20] or salinity stress [50]. We observed a strong relation between Ψtlp and osmotic potential at full turgor across species (R2=0.92; P=0.0022). Nevertheless, it is important to advance in the studies of osmotic function to deal with water and salinity stress, because it has been observed differential responses in terms of osmoregulation depending on the type of the stress [51,52].”
